# Antiprotozoal Aminosteroids from *Pachysandra terminalis*

**DOI:** 10.3390/molecules30051093

**Published:** 2025-02-27

**Authors:** Lizanne Schäfer, Monica Cal, Marcel Kaiser, Pascal Mäser, Thomas J. Schmidt

**Affiliations:** 1University of Münster, Institute of Pharmaceutical Biology and Phytochemistry (IPBP), PharmaCampus Corrensstraße 48, D-48149 Münster, Germany; l_scha57@uni-muenster.de; 2Swiss Tropical and Public Health Institute (Swiss TPH), Kreuzstrasse 2, CH-4123 Allschwil, Switzerland; monica.cal@swisstph.ch (M.C.); marcel.kaiser@swisstph.ch (M.K.); pascal.maeser@swisstph.ch (P.M.); 3University of Basel, Petersplatz 1, CH-4003 Basel, Switzerland

**Keywords:** *Pachysandra terminalis*, Buxaceae, aminosteroids, *Plasmodium falciparum*, *Trypanosoma brucei*, natural products

## Abstract

*Trypanosoma brucei rhodesiense* (*Tbr*) and *Plasmodium falciparum* (*Pf*) are protozoan parasites that cause severe diseases, namely, Human African Trypanosomiasis (HAT) and Malaria. Due to limited treatment options, there is an urgent need for new antiprotozoal drugs. *Pachysandra terminalis* (*P. terminalis*), a plant belonging to the family Buxaceae, is known as a rich source of aminosteroid alkaloids, and a previous study of our working group already showed that the alkaloid-enriched fraction of *P. terminalis* aerial parts showed promising activity against protozoan parasites. In the present study, the alkaloid-enriched fraction obtained from a 75% ethanol extract of aerial parts was separated to isolate a chemically diverse array of *Pachysandra* alkaloids for assessment of their antiprotozoal activity and later structure–activity studies. This work yielded a new megastigmane alkaloid (**1**), 7 new aminosteroids (**2**, **7**, **16**, **17**, **18**, **19**, **20**), along with 10 known aminosteroids (**3**–**5**, **8**, **10**–**15**) and 2 artifacts (**6**, **9**) that were formed during the isolation process. The structures were elucidated by UHPLC/+ESI-QqTOF-MS/MS, as well as extensive 1- and 2D-NMR measurements. The extract and its fractions, as well as the isolated compounds, were tested in vitro against *Tbr* and *Pf,* as well as cytotoxicity against mammalian cells (L6 cell line). The activity (IC_50_ values) of the isolated alkaloids ranged between 0.11 and 26 µM (*Tbr*) and 0.39 and 80 µM (*Pf*). 3α,4α-diapachysanaximine A (**7**) showed the highest activity against *Tbr* (IC_50_ = 0.11 µM) with a selectivity index (SI) of 133 and was also quite active against *Pf* with IC_50_ = 0.63 µM (SI = 23). This compound is, therefore, a promising new antiprotozoal target for further investigations.

## 1. Introduction

Protozoan parasites, such as *Plasmodium falciparum* and *Trypanosoma brucei rhodesiense,* cause severe life-threatening diseases (Malaria and Human African Trypanosomiasis, HAT, respectively). In 2022, 249 million cases of Malaria were reported by the World Health Organization [1]. Against the backdrop of rising resistance to existing antimalarials [2], new drugs are urgently needed. In spite of a decline in annual cases of HAT and the new oral drug fexinidazole in hands [3,4], the search for new compounds with antitrypanosomal activity is still an important goal. Natural products have been found in many instances to show high levels of antiprotozoal activity and could be promising starting points for the development of new antiprotozoal drugs (see, e.g., [5,6]).

In previous studies, we found that various aminosteroids from Apocynaceae [7,8] and Buxaceae [9,10] have very promising activity against African *Trypanosoma* species and *Plasmodium falciparum*, respectively. First, investigations on the mechanism of action indicated that these compounds might act by a mechanism of action unrelated to that of existing drugs [8] and are, hence, of particular interest. It, thus, appears very promising to study this new type of antiprotozoal agents in more detail, i.e., to obtain a wider variety of analogs for structure–activity and mechanism-of-action studies.

*Pachysandra terminalis* Sieb. et Zucc. (*P. terminalis*), also known as “Japanese Spurge” or “Carpet Box”, is an evergreen subshrub belonging to the family of Buxaceae Dumort [11]. It originated in Japan, Korea, and China but is also a very common ornamental plant in Germany (German common names are “Dickmännchen” and “Japanischer Ysander”). Like other plants of the Buxaceae family (e.g., *Buxus sempervirens* L.), *P. terminalis* is known as a rich source of alkaloids, mainly aminosteroids. The first isolations of alkaloids from *P. terminalis* were reported in 1964 [12,13,14]. Subsequently, a great variety of *Pachysandra* alkaloids were obtained, mostly pregnane-type alkaloids, with varying functionalities at positions C-3 and C-20 [15]. Various *Pachysandra* alkaloids have been reported to exert conspicuous biological activity, particularly anticancer effects [16,17,18]. As part of our ongoing research to identify antiprotozoal natural products, various aminotriterpenoids and aminosteroids were isolated from leaves of *Buxus sempervirens* and tested against protozoan parasites, some displaying promising activity [9,10]. Furthermore, the alkaloid-enriched fraction obtained from aerial parts of *P. terminalis* was already reported by our group to possess promising activity against *Trypanosoma brucei rhodesiense* and *Plasmodium falciparum* [19]. *P. terminalis,* therefore, is a promising source of new antiprotozoal aminosteroids.

Therefore, the present study aimed at increasing the chemical diversity of known antiprotozoal aminosteroid alkaloids by isolating such compounds from *P. terminalis* and evaluating them as potential hits or leads with antitrypanosomal and/or antiplasmodial activity. Even though parasites of genera *Trypanosoma* and *Plasmodium* belong to rather remote branches of the Eukaryote tree of life, they may share certain strategies of transmission and survival [20] as well as biochemical targets and susceptibility toward antiprotozoal agents. It will, hence, also be interesting to compare their susceptibility toward *Pachysandra* alkaloids.

## 2. Results and Discussion

### 2.1. Antiprotozoal Activity of the Crude Extract and Its Fractions

In a previous study, our group reported on the antiprotozoal activity of the alkaloid fraction obtained from a dichloromethane (DCM) extract of *P. terminalis* aerial parts [19]. In order to confirm the antiprotozoal activity of the present accession of *P. terminalis*, a DCM extract and a 75% ethanol (EtOH) extract were prepared by maceration of small amounts of dried plant material. Both small-scale extracts were partitioned by acid/base extraction to produce their alkaloid-enriched fractions, which were tested for their antiprotozoal activity against *Plasmodium falciparum* (*Pf*) and *Trypanosoma brucei rhodesiense* (*Tbr*) as well as their cytotoxicity (Cytotox) against L6 rat skeletal myoblasts. The latter has frequently been used as control mammalian cells to assess parasite selectivity (e.g., [7,8,9,10]). The alkaloid fraction of the 75% EtOH extract gave distinctly better results (see Table 1), so this solvent was chosen for the isolation of the active constituents. For the preparative work, 1 kg of dried aerial parts was extracted by percolation with 75% ethanol, and the alkaloid fraction was tested, along with the lipophilic and hydrophilic residual fractions, for antiprotozoal and cytotoxic activity. The test results of this larger-scale extract and its fractions are also reported in Table 1.

A strong increase in antiprotozoal activity was observed for the alkaloid fraction over the crude extract. The antiplasmodial activity increased from the crude extract to the alkaloid fraction from an IC_50_ value of 14 µg/mL to 0.31 µg/mL, while the antitrypanosomal activity increased from 52 µg/mL to 1.8 µg/mL. Because the lipophilic residue also showed a low level of activity, it was also considered in the isolation procedure of pure compounds. The hydrophilic residue showed no activity against the tested parasites and was not further evaluated.

### 2.2. Isolation and Structural Characterization of Alkaloids from Pachysandra terminals

The extraction of 1 kg of dried aerial parts of *P. terminalis* with 75% aqueous ethanol yielded 270 g of crude extract. The following acid/base extraction of the crude extract resulted in 6.3 g of the alkaloid-enriched fraction, which was separated by centrifugal partition chromatography (CPC) using a biphasic system of n-Hexane:Ethylacetate (7:3) (*v*/*v*) and Methanol:H_2_O (7:3) (*v*/*v*). The eluates, guided by their TLC profiles, were combined into 18 CPC fractions. Each fraction was tested for activity against *Pf*, *Tbr,* and cytotoxicity (see Appendix A). From these fractions, individual alkaloids were isolated, as summarized in the isolation scheme shown in Figure 1. Fraction 13 yielded 91.5 mg of pure compound **1**. After recrystallization of CPC fraction 6, a mixture of compounds **19** and **20** (8.5 mg) was obtained. Compounds **2**–**13** and **16**–**18** were isolated by preparative HPLC separation of CPC fractions of the alkaloid fraction, while compounds **14** and **15** were obtained by prep. HPLC from the lipophilic residual fraction. As an example, the enrichment and isolation of compound **7**, guided by UHPLC/+ESI-QqTOF-MS/MS analysis (in the following referred to as LC/MS), is shown in Figure 2.

In total, 1 megastigmane alkaloid (**1**) and 19 pregnane-type aminosteroids (**2**–**20**) were isolated from *P. terminalis* (see Figure 3) with compounds **1**, **2**, **7**, **16**, **17**, **18**, **19**, and **20**, to the best of our knowledge, being described as natural products for the first time. Besides these genuine natural plant metabolites, compounds **6** and **9** were found to be artifacts formed during the isolation process. Compound **13**, although previously described as the reaction product of *N*-methylation of the crude alkaloid mixture from *P. terminalis* [12] as well as a natural compound from *Sarcococca* spp. [21,22] is described here as a natural constituent of *P. terminalis* for the first time. For structural characterization and elucidation, LC/MS, as well as detailed NMR spectroscopic analyses, were carried out. The structures of the previously known aminosteroids desacyl-epipachysamine A (**3**) [14,23], epipachysamine B (**4**) [23,24,25], pactermine A (**5**) [17], pachysamine A (**8**) [26], pachysandrine D (**10**) [14,27], terminaline (**11**) [24,28], *N*-methyl-desacyl-epipachysamine A (**12**) [12,29], sarcodinine (**13**) [12,21,22], epipachysamine A (**14**) [14,29], and spiropachysine (**15**) [30] are summarized in Figure 3. Their spectral data were consistent with the results reported in the cited literature. Also, in all cases of aminosteroids, the mass spectral data, including the characteristic fragmentation pattern, as previously described [19,31], supported the structural assignments. The structures of the new compounds **1**, **2**, **7**, **16**, **17**, **18**, **19**, and **20** are shown in Figure 3 as well. Their spectral data (^1^H- and ^13^C NMR) are listed in Table 2 and Table 3. In all cases, 2D NMR spectra (COSY, HSQC, and HMBC) were evaluated to confirm the structures. The relative configuration of all steroids was confirmed by correlating their ^1^H/^1^H coupling constants as well as nuclear Overhauser effects (NOEs) assessed in ^1^H/^1^H NOESY spectra with dihedral angles and atom distances measured in 3D molecular models generated in the Molecular Operating Environment (MOE).

Based on its molecular formula determined by LC/MS (*m*/*z* 238.2185 [M+H]^+^) as C_15_H_27_NO, compound **1** is not an aminosteroid. The ^1^H NMR spectrum, besides various aliphatic methylene and methine resonances, showed signals belonging to six methyl groups (δ_H10_ = 1.05 ppm, δ_H11_ = 1.02 ppm, δ_H12_ = 1.07 ppm, δ_H13_ = 1.99 ppm, δ_H14/15_ = 2.35 ppm) and one olefinic methine signal (δ_H6_ = 5.82 ppm). The ^13^C NMR spectrum displayed downfield resonances for a carbonyl carbon at δ_C1_ = 199.4 ppm and for one C–C double bond at δ_C5_ = 165.1 ppm, δ_C6_ = 125.3 ppm. By analysis of the 2D spectra (COSY, HSQC, and HMBC), the structure of compound **1** was elucidated as 9-(*N,N*-dimethyl)-5-megastigmene-1-one, a megastigmane alkaloid similar to 9-(*N,N*-dimethyl)-4,7-megastigmendien-3-one previously isolated from *P. terminalis* by Jin et al. [32]. Compound **1**, a new natural product to the best of our knowledge, differs from this known alkaloid by missing the double bond between positions 7 and 8 of the side chain. To evaluate the stereochemistry of **1**, a circular dichroism (CD) spectrum was recorded, which displayed no measurable cotton effect (CE), indicating the presence of a racemate with regard to the stereogenic center C-4, which is in the direct vicinity of the enone-chromophore and would give rise to a CE if present as one stereoisomer. This is not surprising since the proton at C-4, due to conjugation with the α,β-unsaturated keto group, is acidic, so the compound may easily racemize at this position. The configuration at the other chiral center, C-9, was not determined.

Compound **2** was found to be very similar to compound **3** (Desacyl-epipachysamine A), from which its molecular formula, determined by LC/MS as C_24_H_42_N_2_, differs by containing two hydrogens less. This pointed toward the presence of a double bond, which was confirmed by the HSQC and HMBC spectra, which showed an additional downfield methine group resonating at δ_H6_ = 5.53 ppm that belongs to a double bond between positions 5 and 6 (δ_C5_ = 139.5 ppm, δ_C6_ = 124.8 ppm). Compound **2** was, thus, elucidated as 5,6-dehydro-desacyl-epipachysamine A.

The molecular formula of compound **7** was determined by LC/MS as C_31_H_48_N_2_O_2_. Most of the ^1^H and ^13^C NMR signals were common to pachysanaximine A (20 α-dimethylamino-3β-methylamino-4β-benzoyl-5α-pregnane) previously isolated from *Pachysandra axillaris* [33] and *Sarcococca saligna* [34]. Compound **7** showed typical proton signals at δ_H4′/6′_ = 8.09 ppm, δ_H5′_ = 7.66 ppm, and δ_H3′/7′_ = 7.53 ppm belonging to the benzyl group at position C-4 as well as proton signals of the two methylamino groups at position C-3 (δ_H24_ = 2.75 ppm) and C-20 (δ_H22_ = 2.88 ppm and δ_H23_ = 2.70 ppm). The difference from pachysanaxime A is the stereochemistry of compound **7** at positions 3 and 4. The proton signal of H-4 appears as a double doublet (dd), showing one large coupling constant of 12.5 Hz with the axial H-5 and one small coupling constant of 4.3 Hz with an equatorial H-3. The neighboring protons of H-4 (H-3 and H-5) must have different orientations to assume the right dihedral angles matching the measured coupling constants of H-4. The NOESY spectrum of **7** shows a NOE between H-4 and H-19 (which is β-oriented above the plane of the steroid ring system). The benzyl group must, hence, be in the α-position at C-4 since only in this case, H-4 is close enough to the methyl group at position 19 to observe a NOE (see Figure 4). The dihedral angle between H-4 and H-5 was measured with a 3D molecule model in Molecular Operating Environment (MOE) for 177.7 °, which is in agreement with the coupling of 12.5 Hz. The proton at position 3 must, therefore, be β-configurated to assume a smaller dihedral angle of about −52.7° matching the smaller coupling constant with H-4 of 4.3 Hz. Compound **7** was, thus, unambiguously elucidated as 20α-dimethylamino-3α-methylamino-4α-benzoyl-5α-pregnane. Based on the previously isolated compound pachysanaximine A, this new diastereomer is named 3α,4α-diapachysanaximine A.

The molecular formula of compound **16** was determined by LC/MS as C_28_H_48_N_2_O_3_. The ^1^H NMR spectrum of **16** showed three signals in the low field at δ_H2_ = 4.06 ppm, δ_H3_ = 3.84 ppm, and δ_H4_ = 3.74 ppm. These protons could be assigned to the positions 2, 3, and 4 of the steroidal skeleton (ring A) based on their COSY ^1^H-shift-correlation (correlations between H-2 and H-1α, H-1β and H-3, H-3 and H-4, as well as H-4 and H-5 were observed). The location of the senecioylamide group at position 3 was assigned with the chemical shifts of the proton (δ_H3_ = 3.84 ppm) and carbon (δ_C3_ = 54.2 ppm) at position 3, so that positions 2 and 4 must bear hydroxy groups as required by the elemental composition. All three substituents at C-2, 3, and 4 are β-oriented (i.e., the hydrogens are in α-position) as H-3 appeared as a pseudotriplet with a small coupling constant (3.4 Hz) in the ^1^H NMR spectrum. None of these protons showed a NOE with the protons of the β-oriented methyl group at position 19. The protons H-3 and H-4 each showed a NOE to the proton H-5α. These observations are only possible if the three protons H-2, H-3, and H-4 are α-oriented. Based on the previously isolated hookerianamide N from Devkota et al. [35], compound **16** was named 4β-hydroxy-hookerianamide N.

The molecular formula of compound **17** was determined by LC/MS as C_31_H_48_N_2_O_3_. It only contains one more oxygen than compound **7**. Comparing the ^1^H NMR data of compounds **17** and **7**, there is a difference in the proton H-4, which appeared as a double doublet (dd) in compound **7** but as a doublet (d) in compound **17**, suggesting that H-4 has only one direct neighboring proton. This also became evident in the ^1^H/^1^H COSY spectrum, where H-4 only displayed a correlation with H-3. A proton signal for position 5 is not observed, so compound **17** should contain a hydroxy group at this position. This is supported by C-5 appearing as a quaternary carbon signal shifted downfield to δ_C5_ = 77.8 ppm and not as a methine group in the ^13^C NMR and HSQC spectrum. The configuration at C-3 and C-4 was unchanged in comparison with compound **7**, as was deduced from a NOE between H-4 and the protons of the methyl group at position 19 and the coupling constant of H-4 with H-3 of 4.4 Hz, which is only possible if H-3 is β-oriented as well. The structure of compound **17** was, thus, elucidated as 5α-hydroxy-3α,4α-diapachysanaximine A.

The molecular formula of compound **18** was determined by LC/MS as C_30_H_46_N_2_O_3_. The ^1^H and ^13^C NMR spectrum of **18** displayed signals of a dimethylamino group at position 20 (δ_H22_ = 2.88 ppm, δ_C22_ = 43.4 ppm; δ_H23_ = 2.77 ppm, δ_C23_ = 35.8 ppm) and a nicotinamide group at position 3 (δ_C1′_ = 169.6 ppm; δ_C2′_ = 135.7 ppm; δ_H3′/7′_ = 7.49 ppm, δ_C3′/7′_ = 132.8 ppm; δ_H4′/6′_ = 7.87 ppm, δ_C4′/6′_ = 129.6 ppm; δ_H5′_ = 7.56 ppm, δ_C5′_ = 128.3 ppm). The location of this amide group at position 3 was deduced from the chemical shifts (δ_H3_ = 4.05 ppm; δ_C3_ = 55.1 ppm), characteristic of an amide-substituted methine. Moreover, **18** showed two additional proton signals that were shifted downfield (δ_H2_ = 4.19 ppm; δ_H4_ = 3.87 ppm), assigned to the positions 2 and 4 of the steroid skeleton (ring A) based on their COSY ^1^H-shift-correlation (correlations between H-2 and H-1α, H-1β and H-3, H-3 and H-4, as well as H-4 and H-5). At both positions 2 and 4, hydroxy groups are attached to the steroidal skeleton. Based on H-3 appearing as a pseudotriplet (3.5 Hz) in the ^1^H NMR spectrum, the dihedral angle between H-3 and H-4, as well as H-2, must be equal. Because NOEs of H-3 and H-4 with H-5α and between H-4 and H-6α were observed in the NOESY spectrum, the protons at positions 3 and 4 must, hence, be α-oriented, which then also applies to H-2. It should be noted that the spectroscopic features related to the configuration in **18** (coupling constants as well as NOEs) were analogous to compound **16** described above since the two compounds differ only in the nature of their amide group. Based on pachysamine K previously isolated by Sun et al. [36] and reported to have α-oriented substituents at C-2, -3, and -4, the diastereomeric compound **18** was named 2β,3β,4β-diapachysamine K.

After recrystallization of CPC fraction 6, a mixture of compounds **19** and **20** was obtained. The molecular formulas of these compounds were determined by LC/MS as C_23_H_40_N_2_O and C_23_H_38_N_2_O, respectively, differing by two hydrogens. The chemical shifts of the protons of the methyl group 21 were shifted to a significantly lower field, δ_H21(**19/20**)_ = 1.86 ppm, compared to compounds with a mono- or dimethyl amino group at position C-20 (δ_H21_ ≈ 1.33–1.36 ppm), thus indicating that the compounds do not bear an amino group at C-20. Furthermore, the [M+H]^+^ ion of both compounds has a higher intensity than the [M+2H]^2+^ ion, which points in the same direction: aminosteroids with two basic amino groups (at C-3 and C-20) normally show a higher intensity of their [M+2H]^2+^ ion. All other atoms are assembled to the 3β-dimethylaminopregnane (**19**) and pregn-5,6-ene (**20**) skeleton; the only possibility to accommodate the remaining nitrogen atom and hydroxy group in a single substituent at C-20 was the presence of an oxime group. To confirm the oxime substituent at position C-20, a ^15^N NMR spectrum was recorded. The signal of the nitrogen atom in question for compound **19** was recorded at δ_N_ = 349.0 ppm and for **20** at δ_N_ = 362.0 ppm. These values match well with the range given in the literature for oxime nitrogens, which resonate around 330–360 ppm [37] and, thus, confirm the postulated structure of a C20-oxime. Thus, on the grounds of their ^1^H, ^13^C, as well as ^15^N NMR data, the structures of compounds **19** and **20** were elucidated as 3β-dimethylamino-pregnane-20-oxime and 3β-dimethylamino-pregn-5,6-ene-20-oxime, respectively. To the best of our knowledge, oximes are a newly described compound class for *P. terminalis*. Furthermore, no other steroid derivatives with an oxime group at C-20 have previously been isolated as natural products, to the best of our knowledge. The only steroid oximes previously obtained from a natural source, sponges of the genus *Cinachyrella*, bear an oxime group at C-6 [38].

It is noteworthy that oximes **19** and **20** were detectable by LC/MS in the crude extract, so they are very likely genuine plant metabolites. An attempt to further separate the mixture of compounds **19** and **20** by prep. HPLC (RP18, ACN + 0.1% TFA/H_2_O + 0.1% TFA) led to the isolation of compound **9**, as shown in Figure 5. The LC/MS analysis of the isolate showed a mass loss of 15 Da compared to compound **19**. The molecular formula of **9** is determined by LC/MS being C_23_H_39_NO; this difference could be attributed to the loss of NH. Compared to compound **19**, there was a downfield shift of methyl signal 21 (δ_H21_ = 2.11 ppm) in the ^1^H NMR of **9**, as well as a downfield shift of the carbon C-20 (δ_C20_ = 212.3 ppm) in the ^13^C NMR, indicating the presence of a newly formed keto group instead of the oxime. This was supported by all further 2D NMR data, so the structure of **9** was elucidated as 3β-dimethylamino-pregnane-20-one. Compound **9** is not a genuine plant metabolite formed by *P. terminalis*. It was formed from the natural metabolite **19** during the prep. HPLC separation of **19** and **20** due to the acidic conditions that were used for the separation. Under acidic conditions, the oxime–carbonyl equilibrium is shifted to the carbonyl side, resulting in the formation of compound **9** [39]. It is quite straightforward to assume that **20** will have formed the analogous 5,6-dehydro derivative of **9** under these conditions, which, however, was not isolated.

The LC/MS analysis of compound **6** showed the characteristic isotope pattern of a compound containing chlorine. Moreover, the ^1^H NMR of **6**, otherwise very similar to that of **3**, showed a singlet signal for two protons belonging to a CH_2_-group shifted downfield to δ_H1′_ = 5.30 ppm (δ_C1′_ = 69.3 ppm). As this chlorinated methylene group showed no shift-correlation in the ^1^H/^1^H COSY spectrum, it must be connected to the steroidal skeleton through either a quaternary carbon or a hetero atom. The HMBC spectrum of **6** showed a correlation between the chlorinated methylene group and position 3, so that it is, in fact, attached to the ammonium group at C-3, together with two methyl groups (the latter significantly deshielded by about 7 ppm in comparison to the 3-dimethylamino group in, e.g., the parent compound **3**). The structure was, therefore, unambiguously elucidated as *N3*-chloromethyl-desacyl-epipachysamine A. As this compound was not present in the LC/MS chromatogram of the crude extract, it is not a genuine plant metabolite of *P. terminalis*. Compound **6** was probably formed from compound **3** during the acid/base extraction with DCM. The formation of similar products has been described for pyridine derivatives with DCM by Rudine et al. [40].

### 2.3. Antiprotozoal Activity of the Isolated Compounds from Pachysandra terminalis

The antiprotozoal activity of all compounds isolated from the alkaloid enriched fraction and the lipophilic residue was tested against *Pf* and *Tbr* and compared with their cytotoxicity against L6 cells, as described above. The results are reported in Table 4. Since the alkaloids were obtained as mono- or bis-trifluoroacetates, depending on the number of their basic amino groups, these salts were submitted to the biotests, and the molar IC_50_ values were calculated based on the salts’ molecular masses.

All of the isolated compounds showed promising in vitro antiprotozoal activity (IC_50_-values) in the range of 0.11–26 µM for *Tbr* and 0.39–80 µM for *Pf*. Especially compound **7** (3α,4α-diapachysanaximine A) was highly active against *Tbr* with an IC_50_-value of 0.11 µM and had a considerable selectivity index of SI = 133. The similar compound **10**, which differs from **7** only in the substituent at position 4, also showed high activities against *Tbr* (0.95 µM) and *Pf* (1.01 µM). As **7** was the most promising compound isolated from *P. terminalis* in this study, it was selected for an in vivo study against *T. brucei* infection in a mouse model. The isolation of compound **7** in a larger quantity is still in progress.

A comparison of the antitrypanosomal and antiplasmodial activity data with those for cytotoxicity against the mammalian control cells (Figure 6) shows that there is no significant correlation between either of the antiprotozoal activities and cytotoxicity (both correlation coefficients R << 0.5). On the other hand, a slight positive correlation exists between the two antiprotozoal activity sets (R = 0.61). Since related mechanisms of action can be expected to lead to correlation of the resulting activity data, these observations indicate that the mechanism(s) underlying the two different antiprotozoal activities may be more closely related to each other than to that responsible for the (rather weak) mammalian cytotoxicity. It will be very interesting to study the underlying mechanism of action as well as structure–activity relationships of the present set of compounds in comparison with the related aminosteroids isolated from *Holarrhena* [41] and *Buxus* species.

## 3. Materials and Methods

### 3.1. Plant Material

The aerial parts (leaves, stems, and twigs) of *Pachysandra terminalis* were collected in the botanical garden of the University of Münster in March 2022 and identified by T. J. Schmidt. A voucher specimen is deposited at the Institute of Pharmaceutical Biology and Phytochemistry (IPBP), University of Münster (voucher No.: IPBP 883 (TS_PT_02)). The plant material was dried at room temperature for one week and afterward crushed with a mortar.

### 3.2. Extraction of the Plant Material

The dried and crushed aerial parts of *P. terminalis* (1043 g) were extracted through percolation using a glass chromatography tube (92 × 7 cm) as percolator and aqueous ethanol (75%) as extracting solvent. The plant material was soaked with the solvent in the percolator overnight (no flow) and then extracted with 1000 mL of solvent (flow rate approx. 8 mL/min). This procedure was carried out three subsequent times. After evaporating the solvent (rotary evaporator), 270 g of crude extract was obtained. To enrich the alkaloids, the crude extract was further extracted by acid/base extraction. The crude extract (in 15 g portions) was suspended in 500 mL distilled water. The suspension was acidified with hydrochloric acid (aq., 1 M) to a pH of 1 and filtered under reduced pressure. The filtrate was extracted four times with dichloromethane (lipophilic residue; 18.1 g). As the alkaloids are protonated due to the low pH, they are better soluble in water and, therefore, do not transfer to the dichloromethane phase. Afterward, the aqueous phase was basified with sodium hydroxide solution (aq., 2 M) to a pH of 10. The alkaloids were then present as free bases. The aqueous phase was extracted four times with 250 mL dichloromethane. The alkaloids were thereby transferred into the dichloromethane phase. The solvent was evaporated under reduced pressure at 40 °C to obtain the alkaloid fraction (6.3 g). The remaining aqueous phase after evaporation yielded the hydrophilic residue.

### 3.3. Isolation of Alkaloids from Pachysandra terminalis

#### 3.3.1. Isolation of a Megastigmane and Aminosteroids from the Alkaloid Fraction

The alkaloid fraction was further fractionated with centrifugal partition chromatography (CPC) on a CPC-250 (Gilson, Limburg, Germany) chromatograph. For that purpose, the same liquid/liquid phase system was suitable as previously used for *Buxus sempervirens* in our working group [10], which consists of n-Hexane:Ethylacetate (7:3) (*v*/*v*) and Methanol:H_2_O (7:3) (*v*/*v*). The biphasic system was equilibrated overnight in a separatory funnel. The alkaloid fraction (2.5 g in 3 portions of 0.5–1 g) was diluted in 6 mL of the upper phase and 3 mL of the lower phase. The fractionation was carried out in ascending mode (lower phase as the stationary phase) with a flow rate of 3 mL/min and a rotation of 1000 rpm. In each test tube, the eluate was collected for 3 min according to 9 mL. In total, 90 test tubes were collected for the upper phase (elution mode). Afterward, the lower phase was fractionated without rotation and with a flow rate of 10 mL/min. In each test tube, the eluent was collected for 1 min according to 10 mL. In total, 50 test tubes were collected (extrusion mode). The fractions were analyzed by thin layer chromatography (TLC) on TLC plate silica gel 60 F_254_ (Merck KGaA, Darmstadt, Germany) with butan-1-ol:H_2_O:CH_3_COOH (10:3:1) (*v*/*v*/*v*) as the mobile phase. For the visualization, Dragendorff’s spray reagent (bismuth carbonate (0.85 g):H_2_O (40 mL): CH_3_COOH (10 mL):potassium iodide solution (40%, 20 mL)) was used. The fractions of the upper phase (elution mode) were combined into 14 fractions, and the fractions of the lower phase (extrusion mode) were combined into 4 fractions, according to their TLC profiles.

CPC fraction 13 (test tubes 58–64) contained 91.5 mg of compound **1** as a yellow oil. The CPC fractions 2, 6, 7, 8, 10, and 17 were further separated by prep. HPLC on an RP-18 phase (VP 250/21 Nucleodur C-18 HTec with a VP 10/16 Nucleodur C18 HTec pre-column, Macherey-Nagel, Düren, Germany). As the mobile phase, H_2_O (+0.1% TFA, A) and Acetonitrile (+0.1% TFA, B) were used with the following gradient: 0.1 min 5% B, 14.0 min 20% B, 24.0 min 30% B, 30.0 min 32% B, 40.0 min 35% B, 50.0 min 100% B, 60.0 min 100% B. The column oven was set to 40 °C, and the flow rate was 15 mL/min. The fractions were dissolved in MeOH (20 mg/mL), and the injection volume was between 500 and 1000 µL. CPC fraction 2 resulted in the isolation of compound **7** (7.8 mg, t_R_ 33.5 min), **8** (15.0 mg, t_R_ 24.4 min), **10** (2.6 mg, t_R_ 30.7 min), **12** (2.7 mg, t_R_ 22.3 min), **13** (1.5 mg, t_R_ 21.4), and **17** (2.6 mg, t_R_ 30.0 min). Compounds **4** (6.3 mg, t_R_ 26.4 min) and **5** (4.7 mg, t_R_ 25.6 min) were obtained from CPC fraction 8. The separation of CPC fraction 7 yielded compounds **2** (18.1 mg, t_R_ 21.2 min) and **3** (11.4 mg, t_R_ 21.6 min). The separation of CPC fraction 10 resulted in the isolation of compounds **11** (2.5 mg, t_R_ 25.2 min), **16** (1.6 mg, 30.4 min), and **18** (1.1 mg, 33.8 min), and the separation of CPC fraction 17 in the isolation of compound **6** (1 mg, 33.0 min). When trying to separate the two oximes (**19** + **20**) from CPC fraction 6, compound **9** (1.1 mg, 27.5 min) was obtained. The recrystallization of CPC fraction 6 from ethyl acetate at 60 °C yielded a mixture of compounds **19** and **20** (8.5 mg). Compounds **12**, **13**, and **18** were further purified with the same prep-HPLC settings as before but with the following isocratic methods: **12**: 0.1 min 15% B, 40.0 min 15% B, 50.0 min 100% B, 60.0 min 100% B; **13**: 0.1 min 18% B, 40.0 min 18% B, 50.0 min 100% B, 60.0 min 100% B; **18**: 0.1 min 25% B, 40.0 min 25% B, 50.0 min 100% B, 60.0 min 100% B.

#### 3.3.2. Isolation of Aminosteroids from the Lipophilic Residue

Compounds **14** (0.9 mg, t_R_ 36.9 min) and **15** (2.1 mg, t_R_ 42.3 min) were isolated from the lipophilic residual fraction by prep. HPLC on an RP-18 phase (VP 250/21 Nucleodur C-18 HTec with a VP 10/16 Nucleodur C18 HTec pre-column, Macherey-Nagel, Düren, Germany). As a mobile phase, H_2_O (+0.1% TFA, A) and Acetonitrile (+0.1% TFA, B) were used with the following gradient: 0.1 min 5% B, 14.0 min 20% B, 24.0 min 30% B, 30.0 min 35% B, 40.0 min 50% B, 50.0 min 100% B, 60.0 min 100% B. The column oven was set to 40 °C, and the flow rate was 15 mL/min. Compound **15** was further purified with the same prep-HPLC settings as before but with the following isocratic method: 0.1 min 35% B, 40.0 min 35% B, 50.0 min 100% B, 60.0 min 100% B.

### 3.4. Analysis of the Isolated Alkaloids

A Bruker micrOTOF-Q II mass spectrometer (Bruker Daltonics GmbH, Bremen, Germany) coupled to a Dionex Ultimate RS 3000 liquid chromatograph (Thermo Fisher Scientific, Waltham, MA, USA) was used for the UHPLC/+ESI-QqTOF-MS/MS analysis on an RP-18 phase (Dionex Acclaim RSLC 120, Thermo Fisher Scientific, Waltham, MA, USA). The column oven was set to 40 °C, and the flow rate was 0.4 mL/min. As the mobile phase, H_2_O (+0.1% formic acid, A) and Acetonitrile (+0.1% formic acid, B) were used with the gradient that was previously described for *Buxus* alkaloids [42]: −0.880 min 15% B, −0.480 min 15% B, 1.000 min 30% B, 7.000 min 33% B, 9.020 min 50% B, 9.050 min 100% B, 15.000 min 100% B, 15.100 min 15% B, 20.000 min 100% B. The injection volume was 2 µL. The samples were dissolved in the following concentrations: crude extract (10 mg/mL), CPC fractions (1 mg/mL), and pure compounds (0.1 mg/mL). The data were analyzed with DataAnalysis 4.1.

^1^H, ^13^C, and ^15^N NMR spectra, as well as 2D NMR homo- and heteronuclear correlation spectra (^1^H/^1^H COSY and NOESY, ^1^H/^13^C HSQC and HMBC) were recorded in deuterated solvents (CDCl_3_, CD_3_OD (1–10 mg/700 µL)) on Agilent DD2 400 MHz and 600 MHz spectrometers (Agilent, Santa Clara, USA). Chemical shifts are reported in parts per million (ppm) against the solvent residual peak of the undeuterated solvent. Coupling constants are given in Hertz (Hz). The data were analyzed with MestReNova (V.15.0.0-34764).

Circular dichroism (CD) spectra of compounds **1** and **15** were recorded with a Jasco J-815 CD spectrometer (Jasco, Groß-Umstadt, Germany). The sample was dissolved in MeOH (0.1 mg/mL (**1**)/1 mg/mL (**15**)). The measurement was performed in a 1 mm Suprasil Quartz cuvette (Hellma, Müllheim, Germany).

### 3.5. Spectral Data of the Isolated Alkaloids

9-(*N*,*N*-Dimethyl)-5-megastigmen-1-one **(1):** Yellow oil; +ESI-QqTOF-MS (*m*/*z*): 238.2185 [M+H]^+^ (calcd for C_15_H_28_NO^+^: 238.2165). For ^1^H and ^13^C NMR data, see Table 2 and Table 3, respectively. For spectral data (LC/MS, ^1^H NMR, ^13^C NMR, 2D NMR (COSY, HSQC, HMBC), UV, and CD), see Appendix A.

5,6-Dehydro-desacyl-epipachysamine A **(2):** White powder; +ESI-QqTOF-MS (*m*/*z*): 359.3434 [M+H]^+^ (calcd for C_24_H_43_N_2_^+^: 359.3421), 180.1765 [M+2H]^2+^ (calcd for C_24_H_44_N_2_^+^: 180.1752). For ^1^H and ^13^C NMR data, see Table 2 and Table 3, respectively. For spectral data (LC/MS, ^1^H NMR, ^13^C NMR, 2D NMR (COSY, HSQC, HMBC, NOESY)), see Appendix A.

Desacyl-epipachysamine A **(3)**: White powder; +ESI-QqTOF-MS (*m*/*z*): 361.3591 [M+H]^+^ (calcd for C_24_H_45_N_2_^+^: 361.3577), 181.1845 [M+2H]^2+^ (calcd for C_24_H_46_N_2_^2+^: 181.1830);

^1^H and ^13^C NMR data are in agreement with the literature data [14]. For spectral data (LC/MS, ^1^H NMR, ^13^C NMR), see Appendix A.

Epipachysamine B **(4)**: White powder; +ESI-QqTOF-MS (*m*/*z*): 452.3636 [M+H]^+^ (calcd for C_29_H_46_N_3_O^+^: 452.3641), 226.6880 [M+2H]^2+^ (calcd for C_29_H_47_N_3_O^+2+^: 226.6860); ^1^H and ^13^C NMR data are in agreement with the literature data [24]. For spectral data (LC/MS, ^1^H NMR, ^13^C NMR), see Appendix A.

Pactermine A **(5)**: Light yellow gum; +ESI-QqTOF-MS (*m*/*z*): 450.3486 [M+H]^+^ (calcd for C_29_H_44_N_3_O^+^: 450.3484), 225.6803 [M+2H]^2+^ (calcd for C_29_H_45_N_3_O^+2+^: 225.6781); ^1^H and ^13^C NMR data are in agreement with the literature data [17]. For spectral data (LC/MS, ^1^H NMR, ^13^C NMR), see Appendix A.

*N*3-chloromethyl-desacyl-epipachysamine A **(6):** Colorless gum; +ESI-QqTOF-MS (*m*/*z*): 409.3376 [M]^+^ (calcd for C_25_H_46_ClN_2_^+^: 409.3350), 205.1744 [M+H]^2+^ (calcd for C_25_H_47_ClN_2_^2+^: 205.1714). For ^1^H and ^13^C NMR data, see Table 2 and Table 3, respectively. For spectral data (LC/MS, ^1^H NMR, ^13^C NMR, 2D NMR (COSY, HSQC, HMBC, NOESY)), see Appendix A.

3α,4α-Diapachsanaximine A **(7):** yellow gum; +ESI-QqTOF-MS (*m*/*z*): 481.3811 [M+H]^+^ (calcd for C_31_H_49_N_2_O_2_^+^: 481.3794), 241.1964 [M+2H]^2+^ (calcd for C_31_H_50_N_2_O_2_^2+^: 241.1936). For ^1^H and ^13^C NMR data, see Table 2 and Table 3, respectively. For spectral data (LC/MS, ^1^H NMR, ^13^C NMR, 2D NMR (COSY, HSQC, HMBC, NOESY)), see Appendix A.

Pachysamine A **(8):** White powder; +ESI-QqTOF-MS (*m*/*z*): 361.3616 [M+H]^+^ (calcd for C_24_H_45_N_2_^+^: 361.3583), 181.1850 [M+2H]^2+^ (calcd for C_24_H_46_N_2_^2+^: 181.1830); ^1^H and ^13^C NMR data are in agreement with the literature data [26]. For spectral data (LC/MS, ^1^H NMR, ^13^C NMR), see Appendix A.

3β-Dimethylamino-pregnane-20-one **(9):** Colorless gum; +ESI-QqTOF-MS (*m*/*z*): 346.3178 [M+H]^+^ (calcd for C_23_H_40_NO^+^: 346.3110). For ^1^H and ^13^C NMR data, see Table 2 and Table 3, respectively. For spectral data (LC/MS, ^1^H NMR, ^13^C NMR, 2D NMR (COSY, HSQC, HMBC, NOESY)), see Appendix A.

Pachysandrine D (**10**): Yellow gum; +ESI-QqTOF-MS (*m*/*z*): 459.3976 [M+H]^+^ (calcd for C_29_H_51_N_2_O_2_^+^: 459.3951), 230.2053 [M+2H]^2+^ (calcd for C_29_H_52_N_2_O_2_^2+^: 230.2014); ^1^H and ^13^C NMR data are in agreement with the literature data [14,27]. For spectral data (LC/MS, ^1^H NMR, ^13^C NMR), see Appendix A.

Terminaline **(11)**: Yellow gum; +ESI-QqTOF-MS (*m*/*z*): 364.3269 [M+H]^+^ (calcd for C_23_H_42_NO_2_^+^: 364.3216); ^1^H and ^13^C NMR data are in agreement with the literature data [24,28]. For spectral data (LC/MS, ^1^H NMR, ^13^C NMR), see Appendix A.

*N*-methyl-desacyl-epipachysamine A **(12):** Light yellow gum; +ESI-QqTOF-MS (*m*/*z*): 375.3774 [M+H]^+^ (calcd for C_25_H_47_N_2_^+^: 375.3739), 188.1943 [M+2H]^2+^ (calcd for C_25_H_48_N_2_^2+^: 188.1909); ^1^H and ^13^C NMR data are in agreement with the literature data [12,29]. For spectral data (LC/MS, ^1^H NMR, ^13^C NMR), see Appendix A.

Sarcodinine **(13):** Colorless gum; +ESI-QqTOF-MS (*m*/*z*): 373.3659 [M+H]^+^ (calcd for C_25_H_45_N_2_^+^: 373.3583), 187.1888 [M+2H]^2+^ (calcd for C_25_H_46_N_2_^2+^: 187.1830); ^1^H and ^13^C NMR data are in agreement with the literature data [12]. For spectral data (LC/MS, ^1^H NMR, ^13^C NMR), see Appendix A.

Epipachysamine A **(14):** Colorless gum; +ESI-QqTOF-MS (*m*/*z*): 403.3713 [M+H]^+^ (calcd for C_26_H_47_N_2_O^+^: 403.3688), 202.1913 [M+2H]^2+^ (calcd for C_26_H_48_N_2_O^2+^: 202.1883); ^1^H and ^13^C NMR data are in agreement with the literature data [14,29]. For spectral data (LC/MS, ^1^H NMR, ^13^C NMR), see Appendix A.

Spiropachysine **(15):** Colorless gum; +ESI-QqTOF-MS (*m*/*z*): 463.3683 [M+H]^+^ (calcd for C_31_H_47_N_2_O^+^: 463.3688), 232.1867 [M+2H]^2+^ (calcd for C_31_H_48_N_2_O ^2+^: 232.1883); ^1^H and ^13^C NMR data, as well as CD spectrum (Appendix A), are in agreement with the literature data [30]. For spectral data (LC/MS, ^1^H NMR, ^13^C NMR, UV, and CD), see Appendix A.

4β-Hydroxy-hookerianamide N **(16):** Light yellow gum; +ESI-QqTOF-MS (*m*/*z*): 461.3815 [M+H]^+^ (calcd for C_28_H_49_N_2_O_3_^+^: 461.3743), 231.1959 [M+2H]^2+^ (calcd for C_28_H_50_N_2_O_3_^2+^: 231.1911). For ^1^H and ^13^C NMR data, see Table 2 and Table 3, respectively. For spectral data (LC/MS, ^1^H NMR, ^13^C NMR, 2D NMR (COSY, HSQC, HMBC, NOESY)), see Appendix A.

5α-Hydroxy-3α,4α-diapachysanaximine A **(17):** Light yellow gum; +ESI-QqTOF-MS (*m*/*z*): 497.3757 [M+H]^+^ (calcd for C_31_H_49_N_2_O_3_^+^: 497.3743), 249.1934 [M+2H]^2+^ (calcd for C_31_H_50_N_2_O_3_^2+^: 249.1911). For ^1^H and ^13^C NMR data, see Table 2 and Table 3, respectively. For spectral data (LC/MS, ^1^H NMR, ^13^C NMR, 2D NMR (COSY, HSQC, HMBC, NOESY)), see Appendix A.

2β,3β,4β-Diapachysamine K **(18):** White powder; +ESI-QqTOF-MS (*m*/*z*): 483.3640 [M+H]^+^ (calcd for C_30_H_47_N_2_O_3_^+^: 483.3587), 242.1888 [M+2H]^2+^ (calcd for C_30_H_48_N_2_O_3_
^2+^: 242.1832). For ^1^H and ^13^C NMR data, see Table 2 and Table 3, respectively. For spectral data (LC/MS, ^1^H NMR, ^13^C NMR, 2D NMR (COSY, HSQC, HMBC, NOESY)), see Appendix A.

3β-Dimethylamino-pregnane-20-oxime **(19):** Yellow powder; +ESI-QqTOF-MS (*m*/*z*): 361.3225 [M+H]^+^ (calcd for C_23_H_41_N_2_O^+^: 361.3219), 181.1667 [M+2H]^2+^ (calcd for C_23_H_42_N_2_O^2+^: 181.1649). For ^1^H and ^13^C NMR data, see Table 2 and Table 3, respectively.

3β-Dimethylamino-pregn-5,6-ene-20-oxime **(20):** Yellow powder; +ESI-QqTOF-MS (*m*/*z*): 359.3067 [M+H]^+^ (calcd for C_23_H_39_N_2_O^+^: 359.3062), 180.1586 [M+2H]^2+^ (calcd for C_23_H_40_N_2_O^2+^: 180.1570). For ^1^H and ^13^C NMR data, see Table 2 and Table 3, respectively. For spectral data (LC/MS, ^1^H NMR, ^13^C NMR, ^15^N NMR, 2D NMR (COSY, HSQC, HMBC, NOESY)), see Appendix A.

### 3.6. Biological Activity Assays

The in vitro assays were performed using well-established standard protocols in the laboratories of Swiss TPH. They are summarized as follows:

*Trypanosoma brucei rhodesiense* (*Tbr*): The STIB900 strain is a derivative of the STIB704 strain isolated from a patient in Ifakara, Tanzania, in 1982 [43]. The bloodstream forms were cultivated as axenic culture in HMI-9 medium, supplemented with 15% heat-inactivated horse serum. Cultures were maintained at 37 °C in an atmosphere of 5% CO_2_. Test compounds were dissolved in DMSO (10 mg/mL), and serial drug dilutions of eleven 3-fold dilution steps covering a range from 100 to 0.002 μg/mL were prepared in a 96-well plate. Bloodstream-form of *T. b rhodesiense* strain STIB900 (2 × 10^3^/well) was added into the wells. After 68 h incubation, 10 μL of resazurin solution (12.5 mg resazurin in 100 mL 1xPBS) was added per well, and plates were incubated for an additional 4 h. Subsequently, the plates were read with a Spectramax Gemini EM microplate fluorometer (Molecular Devices, San Jose, CA, USA) using an excitation wavelength of 536 nm and an emission wavelength of 588 nm. Data were treated according to [44] (see below). Melarsoprol was used as a positive control.

*Plasmodium falciparum* (*Pf*): Tests were performed against erythrocytic stages of the drug-sensitive NF54 strain. The NF54 strain was obtained from F. Hoffmann-La Roche Limited (Basel, Switzerland). The strain NF54 was originally derived from a patient living near Schiphol Airport, Amsterdam, who had never left the Netherlands [45]. The assay was performed using a 3H-hypoxanthine incorporation assay [46,47]. Serial compound dilutions were prepared with the medium as indicated above and then added in 96-well plates to parasite cultures incubated in the medium previously described [48,49], consisting of RPMI 1640 supplemented with 0.5% ALBUMAX^®^ II, 25 mM Hepes, 25 mM NaHCO_3_ (pH 7.3), 0.36 mM hypoxanthine, and 100 μg/mL neomycin. The plates were incubated in a humidified atmosphere at 37 °C, 4% CO_2_, 3% O_2_, and 93% N_2_. After 48 h, 0.25 μCi of 3H-hypoxanthine was added to each well. Incubation was continued under identical conditions for a further 24 h. The plates were then harvested using a Betaplate™ cell harvester (Wallac, Zurich, Switzerland). The red blood cells were transferred onto a glass fiber filter and washed with distilled H_2_O. After drying, filters were inserted into a plastic foil with 10 mL of scintillation fluid and counted with a Betaplate™ liquid scintillation counter (Wallac, Zurich, Switzerland). Data were treated according to [44] (see below). The positive control was chloroquine.

*Cytotoxicity against L-6 rat skeletal myoblasts:* The L6 rat skeletal myoblast cell line used for cytotoxicity tests was obtained from the American Type Culture Collection (ATCC, Manassas, VA, USA) as ATCC-CRL-1458 [50]. The cells were cultivated in RPMI 1640 medium supplemented with 1% L-glutamine (200mM) and 10% fetal bovine serum. Cultures were maintained at 37 °C in an atmosphere of 5% CO_2_. Assays were performed in 96-well plates, each well containing RPMI 1640 medium supplemented with 1% L-glutamine (200 mM), 10% fetal bovine serum, and 2 × 10^3^ L-6 cells/well. Plates were incubated at 37 °C under a 5% CO_2_ atmosphere for 24 h. Compounds were dissolved in DMSO (10 mg/mL). After 24 h, serial drug dilutions of eleven 3-fold dilution steps covering a range from 100 to 0.002 μg/mL were prepared, and the plates were incubated at 37 °C under 5% CO_2_ for 70 h. After 70 h incubation, 10 μL of resazurin solution (12.5 mg resazurin in 100 mL 1xPBS) was added per well, and plates were incubated for an additional 2 h. After that, the plates were read with a Spectramax Gemini EM microplate fluorometer (Molecular Devices, San Jose, CA, USA) using an excitation wavelength of 536 nm and an emission wavelength of 588 nm. Data were treated according to [44] (see below). Podophyllotoxin served as a positive control.

*Bioassay data and statistical methods:* In all cases, the readout data were plotted in Microsoft Excel. Half-maximal inhibitory concentrations (IC_50_ values) were calculated from the sigmoidal dose-response curves by linear regression [44]. Two independent replicates of the assay were performed in all cases, and the results were expressed as arithmetic mean ± deviation from the mean.

## 4. Conclusions

As already expected on the background of our previous study [19], the aerial parts of *P. terminalis* yielded a rich variety of aminosteroids with antiplasmodial and antitrypanosomal activity. In the present study, 20 alkaloids were isolated from the very complex 75% ethanolic extract, confirming that this plant is a great source of new antiprotozoal alkaloids. Among these, 10 compounds were previously undescribed and could be elucidated by their NMR and LC/MS data as eight new native *Pachysandra* alkaloids and two aminosteroids representing artifacts formed during the isolation procedure. The alkaloid-enriched fraction of *P. terminalis,* as well as the isolated compounds, showed prominent activity against *Plasmodium falciparum* and *Trypanosoma brucei rhodesiense* with 3α,4α-diapachysanaximine A (**7**) being the most active compound obtained from this plant extract, with IC_50_ values against both parasites in the lower µM range. Aminosteroids derived from *P. terminalis* thus represent interesting antiparasitic hits to be further investigated. The isolation of 3α,4α-diapachysanaximine A in larger quantities for in vivo assays is in progress. Investigations on structure–activity relationships of the isolated *Pachysandra* alkaloids in comparison with those obtained from *Buxus* and *Holarrhena* species are underway.

## Figures and Tables

**Figure 1 molecules-30-01093-f001:**
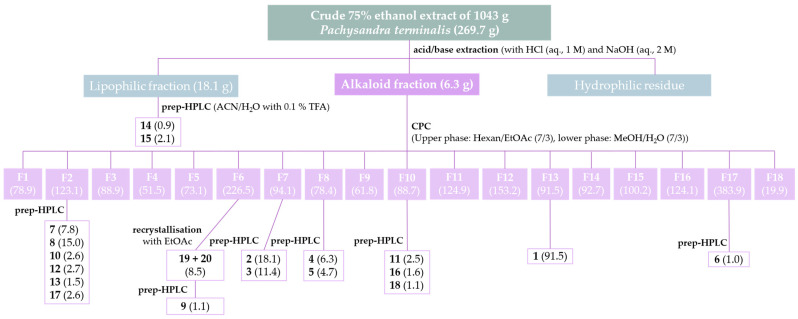
Isolation scheme for alkaloids from aerial parts of *Pachysandra terminalis*. The numbers in brackets represent the yields in mg.

**Figure 2 molecules-30-01093-f002:**
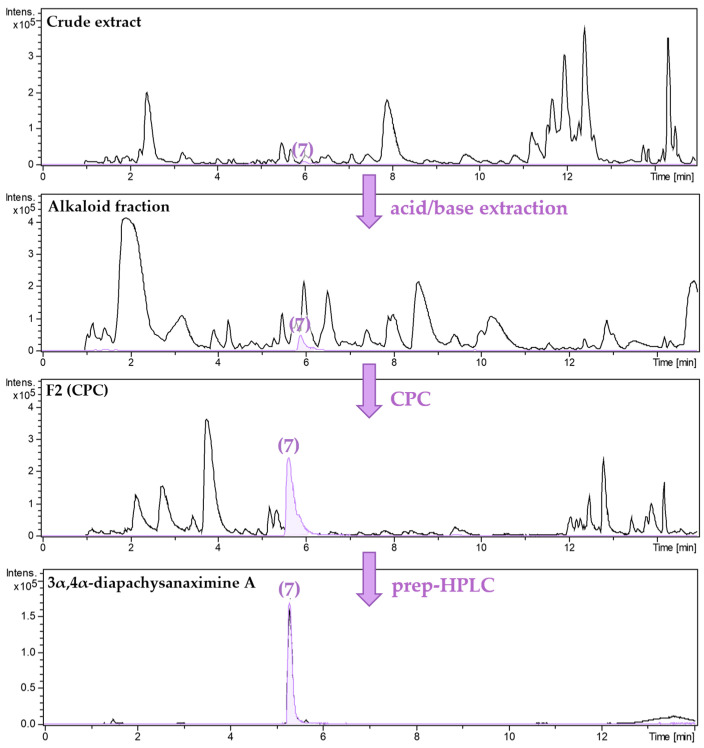
UHPLC/+ESI-QqTOF-MS chromatograms of the isolation of 3α,4α-diapachysanaximine A **(7)** from the crude extract of *Pachysandra terminalis*. Base peak chromatogram (*m*/*z* 200–1000) in black and extracted ion chromatogram of the [M+2H]^2+^ ion (*m*/*z* 241.1964) of **7** in purple.

**Figure 3 molecules-30-01093-f003:**
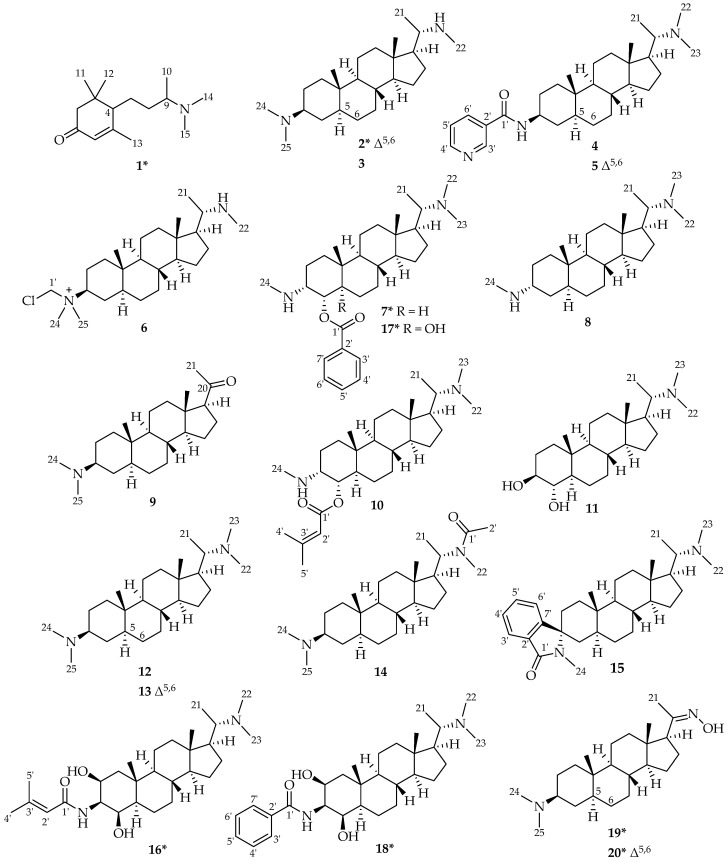
Chemical structures of alkaloids isolated from *Pachysandra terminalis*. * The compounds marked with an asterisk are, to the best of our knowledge, new natural compounds. Compounds **6** and **9** were found to be artifacts formed during the isolation process.

**Figure 4 molecules-30-01093-f004:**
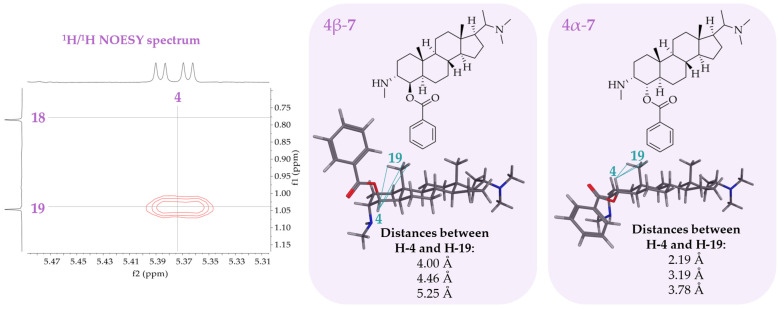
^1^H/^1^H NOESY spectrum in the region of H-4 of compound **7**. Molecular models of the two conformers of **7** with the substituent at position C-4 in α- and β-orientation. The distances between the proton H-4 and the protons of the methyl group at C-19 were measured.

**Figure 5 molecules-30-01093-f005:**
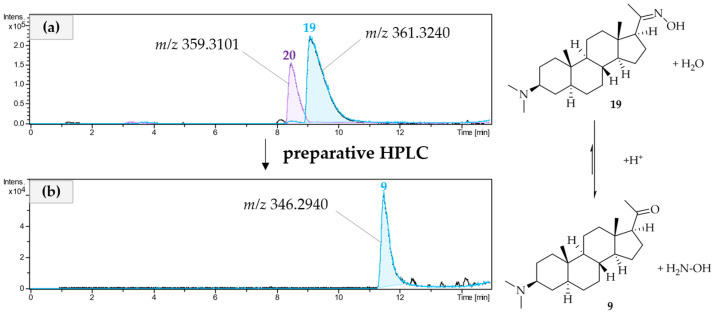
UHPLC/+ESI-QqTOF-MS chromatogram of CPC fraction 6 before preparative HPLC (**a**). UHPLC/+ESI-QqTOF-MS chromatogram of compound **9** that was isolated from CPC fraction 6 (**b**). The reaction from the Oxime (**19**) in CPC fraction 6 to the corresponding ketone (**9**) is shown on the right side.

**Figure 6 molecules-30-01093-f006:**
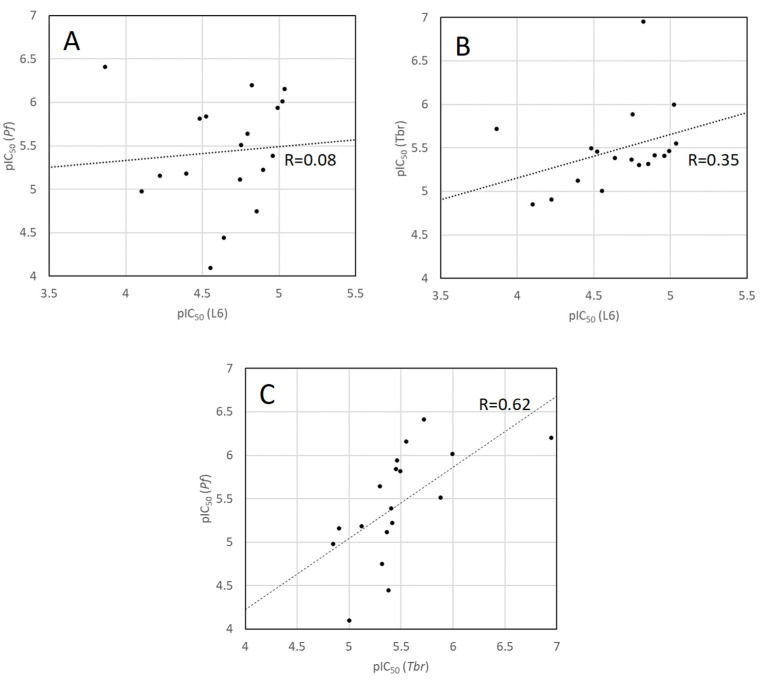
Correlation plots of the biological activity data [pIC_50_ = −log(IC_50_ in mol/L)] of the isolated aminosteroids showing the antiplasmodial and antitrypanosomal activity plotted vs. cytotoxicity ((**A**,**B**), respectively) and versus each other (**C**).

**Table 1 molecules-30-01093-t001:** IC_50_-values (in µg/mL) of the in vitro antiprotozoal (*Plasmodium falciparum*, *Pf*, *Trypanosoma brucei rhodesiense*, *Tbr*) as well as cytotoxic activity (L6 rat skeletal myoblasts) of crude extracts from aerial parts of *Pachysandra terminalis* and the fractions of the acid/base extraction. Data represent arithmetic means of two independent determinations ± their deviation from the mean. SI: Selectivity indices: IC_50_(Cytotox)/IC_50_(*Pf* or *Tbr*).

	*Pf*	*Tbr*	Cytotox	SI (*Pf*)	SI (*Tbr*)
Small scale extracts					
DCM extract	8.5 ± 0.1	5.6 ± 0.6	31 ± 8	3.6	5.5
Alkaloid fraction (DCM)	0.96 ± 0.04	6.07 ± 0.02	17 ± 1	17.3	2.7
75% EtOH extract	9.3 ± 0.5	13.15 ± 0.05	45 ± 2	4.8	3.4
Alkaloid fraction (EtOH)	0.33 ± 0.01	0.85 ± 0.37	14 ± 1	41.2	16.0
Large-scale extracts and fractions					
75% EtOH extract	14 ± 1	52 ± 11	76 ± 3	5.6	1.5
Alkaloid fraction	0.31 ± 0.03	1.8 ± 0.4	13 ± 3	41.0	6.9
Lipophilic residue	2.5 ± 0.2	2.0 ± 0.2	15 ± 1	6.0	7.4
Hydrophilic residue	>50	>100	>100	-	-
Positive controls					
Chloroquine	0.002 ± 0.000				
Melarsoprol		0.007 ± 0.002			
Podophyllotoxin			0.009 ± 0.001		

**Table 2 molecules-30-01093-t002:** ^1^H NMR data of compounds **1**, **2**, **6**, **7**, **9**, **16**, **17**, **18**, **19**, and **20** (600 MHz).

	δ_H_ [ppm], mult., J [Hz]									
Pos.	1 ^a^	2 ^b^	6 ^b^	7 ^b^	9 ^b^	16 ^b^	17 ^b^	18 ^b^	19 ^a^	20 ^a^
1	-	2.05, m 1.20, m	1.95, m 1.17, m	1.73, m 1.27, m	1.92, m 1.12, ddd (dt), 13.5, 4.1	2.12, m 1.31, m	1.73, m 1.43, m	2.17, dd, 14.4, 3.1 1.36, m	1.79, m 0.97, m	1.89, m 1.07, m
2	2.40, d, 17.3 2.05, d, 17.3	1.99, m 1.74, m	2.03, m 1.78, m	2.12, m 2.11, m	1.91, m 1.63, m	4.06, dd, 3.3, 1.4	2.12, m 2.16, m	4.19, d, 3.4	1.82, m 1.50, m	1.83, m 1.49, m
3	-	3.08, dddd (tt), 12.2, 4.1	3.69, dddd (tt), 12.3, 3.9	3.68, m	3.19, dddd (tt), 13.2, 4.3	3.84, dd (t), 3.4	3.63, m	4.05, dd (t), 3.5	2.53, m	2.25, m
4	1.87, m	2.49, dt 12.6, 12.5, 2.6 2.45, ddd, 12.8, 4.6, 2.6	1.76, m 1.67, m	5.37, dd, 12.5/4.3	1.67, m 1.47, m	3.74, m	5.46, d, 4.4	3.87, br s	1.57, m 1.35, m	2.25, m
5	-	-	1.31, m	1.73, m	1.25, m	1.25, m	-	1.31, m	1.11, m	-
6	5.82, s	5.53, dd, 4.9, 2.4	1.37, m, 2H	1.79, m 1.32, m	1.38, m, 2H	1.86, m 1.41, m,	1.61, m 1.36, m	1.89, m 1.41, m	1.29, m 1.25, m	5.34, m
7	1.71, m 1.48, m	2.06, m 1.63, m	1.76, m 1.02, m	1.81, m 1.00, m	1.74, dd 13.3/3.3 1.00, m	1.80, m 1.05, m	1.68, m 1.61, m	1.82, m 1.05, m	1.68, m 0.89, m	1.99, m 1.55, m
8	1.78, m 1.41, m	1.56, m	1.42, m	1.53, m	1.44, m	1.51, m	1.53, m	1.52, m	1.33, m	1.45, m
9	2.68, br s	1.05, m	0.78, m	0.98, m	0.79, ddd, 12.2, 10.5, 4.1	0.73, m	1.17, m	0.76, m	0.67, m	0.97, m
10	1.05, d, 3H	-	-	-	-	-	-	-	-	-
11	1.02, s, 3H	1.62, m 1.57, m	1.59, m 1.36, m	1.62, m 1.40, m	1.64, m 1.36, m	1.55, m 1.38, m	1.51, m 1.35, m	1.57, m 1.40, m	1.26, m 1.54, m	1.43, m 1.29, m
12	1.07, s, 3H	2.03, m 1.31, m	1.98, m 1.27, m	1.99, m 1.34, m	2.04, m 1.46, m	1.96, m 1.29, m	1.98, m 1.34, m	1.98, m 1.32, m	1.85, m 1.26, m	1.85, m 1.26, m
13	1.99, s, 3H	-	-	-	-	-	-	-	-	-
14	2.35, s, 6H	1.17, m	1.18, m	1.24, m	1.23, m	1.22, m	1.26, m	1.23, m	1.10, m	1.04, m
15		1.78, m 1.30, m	1.77, m 1.30, m	1.64, m 1.29, m	1.70, m 1.22, m	1.82, m 1.31, m	1.76, m 1.27, m	1.80, m 1.30, m	1.65, m 1.16, m	1.65, m 1.16, m
16		1.96, m 1.54, m	1.92, m 1.49, m	1.90, m 1.55, m	2.13, m 1.66, m	1.89, m 1.51, m	1.90, m 1.55, m	1.91, m 1.52, m	2.08, m 1.65, m	2.08, m 1.65, m
17		1.59, m	1.58, m	1.71, m	2.63, t, 9.1	1.67, m	1.67, m	1.68, m	2.20, m	2.08, m
18		0.79, s, 3H	0.77, s, 3H	0.79, s, 3H	0.61, s, 3H	0.77, s, 3H	0.78, s, 3H	0.78, s, 3H	0.61, s, 3H	0.64, s, 3H
19		1.07, s, 3H	0.90, s, 3H	1.05, s, 3H	0.88, s, 3H	1.27, s, 3H	1.16, s, 3H	1.30, s, 3H	0.79, s, 3H	0.99, s, 3H
20		3.23, m	3.21, m	3.38, m	-	3.35, m	3.36, m	3.36, m	-	-
21		1.36, d, 6.6, 3H	1.35, d, 6.6 3H	1.33, d, 6.6, 3H	2.11, s, 3H	1.33, d, 6.7, 3H	1.33, d, 6.6, 3H	1.33, d, 6.6, 3H	1.86, s, 6.6, 3H	1.86, s, 6.6, 3H
22		2.65, s, 3H	2.65, s, 3H	2.88, s, 3H		2.88, s, 3H	2.88, s, 3H	2.88, s, 3H		
23				2.70, s, 3H		2.70, s, 3H	2.70, s, 3H	2.70, s, 3H		
24		2.88, s, 6H	3.16, s, 6H	2.75, s, 3H	2.84, s, 3H		2.69, s, 3H		2.47, s (broad), 6H	2.38, s (broad), 6H
25										
1′			5.30, s	-		-	-	-		
2′				-		5.83, sep, 1.3	-	-		
3′/7′				7.53, m		-	8.20, m	7.87, dd, 7.3, 1.0, 2H		
4′/6′				8.09, m		1.88, d, 1.3, 3H	7.53, m	7.49, d, 7.8, 2H		
5′				7.66, m		2.13, d, 1.3, 3H	7.67, m	7.56, t, 7.8		

^a^ recorded in CDCl_3_; ^b^ recorded in CD_3_OD.

**Table 3 molecules-30-01093-t003:** ^13^C NMR data of compounds **1**, **2**, **6**, **7**, **9**, **16**, **17**, **18**, **19**, and **20** (151 MHz). Multiplicities according to ^1^H/^13^C HSQC.

	δC [ppm]									
Pos.	1 ^a^	2 ^b^	6 ^b^	7 ^b^	9 ^b^	16 ^b^	17 ^b^	18 ^b^	19 ^a^	20 ^a^
1	199.4, Cq	38.4, CH_2_	38.4, CH_2_	32.0, CH_2_	37.9, CH_2_	45.8, CH_2_	26.6, CH_2_	45.7, CH_2_	37.5, CH_2_	38.3, CH_2_
2	47.4, CH_2_	23.9, CH_2_	22.4, CH_2_	21.8, CH_2_	23.8, CH_2_	72.0, CH	21.9, CH_2_	71.5, CH	24.8, CH_2_	23.7, CH_2_
3	36.4, Cq	67.4, CH	73.6, CH	60.1, CH	66.9, CH	54.2, CH	60.2, CH	55.1, CH	64.8, CH	65.2, CH
4	51.4, CH	33.9, CH_2_	28.7, CH_2_	72.9, CH	29.9, CH_2_	76.4, CH	73.5, CH	75.8, CH	30.1, CH_2_	34.8, CH_2_
5	165.1, Cq	139.5, Cq	46.8, CH	45.9, CH	46.3, CH	51.6, CH	77.8, Cq	51.4, CH	45.7, CH	141.3, Cq
6	125.3, CH	124.8, CH	29.6, CH_2_	25.1, CH_2_	29.6, CH_2_	27.0, CH_2_	21.8, CH_2_	27.0, CH_2_	28.9, CH_2_	121.4, CH
7	27.2, CH_2_	32.8, CH_2_	32.8, CH_2_	32.1, CH_2_	33.0, CH_2_	33.2, CH_2_	28.6, CH_2_	33.3, CH_2_	32.0, CH_2_	32.2, CH_2_
8	33.4, CH_2_	32.9, CH	36.5, CH	36.0, CH	36.6, CH	36.1, CH	35.2, CH	36.1, CH	35.8, CH	32.0, CH
9	60.3, CH	51.1, CH	54.9, CH	55.0, CH	55.2, CH	57.6, CH	46.0, CH	57.5, CH	54.4, CH	50.4, CH
10	12.9, CH_3_	37.7, Cq	36.3, Cq	38.9, Cq	36.7, Cq	36.1, Cq	42.3, Cq	36.1, Cq	35.9, Cq	37.0, Cq
11	29.0, CH_3_	21.9, CH_2_	22.1, CH_2_	21.8, CH_2_	22.3, CH_2_	21.5, CH_2_	26.1, CH_2_	21.5, CH_2_	21.3, CH_2_	21.1, CH_2_
12	27.3, CH_3_	40.1, CH_2_	40.4, CH_2_	40.3, CH_2_	40.0, CH_2_	40.4, CH_2_	40.4, CH_2_	40.4, CH_2_	39.0, CH_2_	38.8, CH_2_
13	24.7, CH_3_	43.8, Cq	44.0, Cq	44.2, Cq	45.3, Cq	44.3, Cq	44.2, Cq	44.3, Cq	44.1, Cq	43.9, Cq
14	40.4, CH_3_	57.5, CH	57.2, CH	57.2, CH	57.7, CH	57.3, CH	57.0, CH	57.3, CH	56.0, CH	56.3, CH
15		25.2, CH_2_	25.1, CH_2_	23.6, CH_2_	25.4, CH_2_	25.2, CH_2_	25.0, CH_2_	25.2, CH_2_	24.3, CH_2_	24.4, CH_2_
16		27.2, CH_2_	27.2, CH_2_	26.7, CH_2_	23.6, CH_2_	26.8, CH_2_	26.8, CH_2_	26.8, CH_2_	23.2, CH_2_	23.2, CH_2_
17		54.1, CH	54.2, CH	53.1, CH	64.7, CH	53.1, CH	53.1, CH	53.2, CH	57.0, CH	56.9, CH
18		12.2, CH_3_	12.4, CH_3_	12.5, CH_3_	13.8, CH_3_	12.5, CH_3_	12.5, CH_3_	12.5, CH_3_	13.5, CH_3_	13.3, CH_3_
19		19.5, CH_3_	12.5, CH_3_	13.0, CH_3_	12.5 CH_3_	17.4, CH_3_	15.9, CH_3_	17.3, CH_3_	12.5, CH_3_	19.6, CH_3_
20		59.7, CH	59.7, CH	67.0, CH	212.3, Cq	67.1, CH	67.0, CH	67.1, CH	158.9, Cq	158.8, Cq
21		15.9, CH_3_	15.9, CH_3_	12.0, CH_3_	31.6, CH_3_	11.9, CH_3_	12.0, CH_3_	12.0, CH_3_	15.2, CH_3_	15.2, CH_3_
22		29.5, CH_3_	29.5, CH_3_	43.4, CH_3_		43.4, CH_3_	43.4, CH_3_	43.4, CH_3_		
23				35.8, CH_3_		35.8, CH_3_	35.8, CH_3_	35.8, CH_3_		
24		40.5, CH_3_	47.6, CH_3_	33.2, CH_3_	40.4, CH_3_		33.5, CH_3_		40.9, CH_3_	40.4, CH_3_
25										
1′			69.3, CH_2_	166.9, Cq		162.2, Cq	167.1, Cq	169.6, Cq		
2′				130.5, Cq		119.7, CH	130.7, Cq	135.7, Cq		
3′/7′				129.8, CH		139.7, Cq	131.2, CH	129.6, CH		
4′/6′				130.9, CH		27.2, CH_3_	129.7, CH	132.8, CH		
5′				134.9, CH		20.0, CH_3_	134.94, CH	128.30, CH		

^a^ recorded in CDCl_3_; ^b^ recorded in CD_3_OD.

**Table 4 molecules-30-01093-t004:** IC_50_-values of the in vitro antiprotozoal and cytotoxic activity of the pure compounds isolated from *Pachysandra terminalis*. Compounds isolated by prep. HPLCs were tested as the mono ^a^- or bis ^b^-trifluoroacetates. The values in brackets are IC_50_-values in µM that were calculated with the molecular masses of the corresponding salts ^a,b^. SI: Selectivity indices: IC_50_(Cytotox)/IC_50_(*Pf* or *Tbr*).

Compound	*Pf* [µg/mL]	*Tbr* [µg/mL]	Cytotox [µg/mL]	SI (*Pf*)	SI(*Tbr*)
**1**	0.93 ± 0.07 (3.91 µM)	6.2 ± 0.9 (26.2 µM)	47 ± 1 (198 µM)	51	8
**2** ^b^	0.898 ± 0.002 (1.53 µM)	1.9 ± 0.2 (3.2 µM)	19 ± 5 (33 µM)	21	10
**3 ^b^**	0.85 ± 0.03 (1.44 µM)	2.0 ± 0.2 (3.5 µM)	18 ± 3 (30 µM)	21	7
**4 ^b^**	2.3 ± 0.2 (3.4 µM)	1.6 ± 0.5 (2.4 µM)	5.2 ± 0.3 (7.7 µM)	2	3
**5 ^b^**	0.91 ± 0.02 (1.34 µM)	1.93 ± 0.06 (2.85 µM)	5.8 ± 0.4 (8.6 µM)	6	3
**6 ^b^**	0.25 ± 0.03 (0.39 µM)	1.2 ± 0.4 (1.9 µM)	87 ± 13 (136 µM)	346	74
**7 ^b^**	0.45 ± 0.09 (0.63 µM)	0.079 ± 0.001 (0.112 µM)	10 ± 4 (15 µM)	23	133
**8 ^b^**	1.82 ± 0.06 (3.09 µM)	0.8 ± 0.1 (1.3 µM)	10.4 ± 0.8 (17.7 µM)	6	13
**9 ^a^**	8 ± 2 (18 µM)	2.21 ± 0.06 (4.80 µM)	6 ± 2 (14 µM)	0.8	3
**10 ^b^**	0.66 ± 0.13 (0.97 µM)	0.69 ± 0.03 (1.01 µM)	6.5 ± 0.7 (9.5 µM)	10	9
**11 ^a^**	3 ± 1 (7 µM)	5.9 ± 0.3 (12.4 µM)	28 ± 13 (60 µM)	8	5
**12 ^b^**	2.5 ± 0.3 (4.1 µM)	2.3 ± 0.2 (3.9 µM)	7 ± 2 (11 µM)	3	3
**13 ^b^**	4.6 ± 0.7 (7.7 µM)	2.6 ± 0.4 (4.3 µM)	11 ± 3 (18 µM)	2	4
**14 ^a^**	18 ± 1 (36 µM)	2.14 ± 0.05 (4.14 µM)	12 ± 3 (23 µM)	0.6	5
**15 ^a^**	3 ± 1 (6 µM)	2.21 ± 0.04 (3.83 µM)	7.3 ± 0.7 (12.7 µM)	2	3
**16 ^a^**	6.1 ± 0.5 (10.6 µM)	8.1 ± 0.6 (14.1 µM)	45 ± 1 (79 µM)	7	6
**17 ^b^**	1.6 ± 0.4 (2.3 µM)	3 ± 2 (5 µM)	12 ± 1 (16 µM)	7	3
**18 ^a^**	47 ± 7 (80 µM)	5.9 ± 0.4 (9.9 µM)	17 ± 2 (28 µM)	0.4	3
**19** ^c^ **+ 20 ^c^**	2.38 ± 0.11 (6.61 µM)	2.71 ± 0.67 (7.52 µM)	14.51 ± 0.31 (40.27 µM)	6	5
Trifluoroacetate	>100	>100	>100	-	-
Chloroquine	0.002 ± 0.000				
Melarsoprol		0.007 ± 0.002			
Podophyllotoxin			0.009		

^a^ mono-trifluroracetate; ^b^ bis-trifluoroacetate. ^c^ Compounds **19** and **20** were isolated as mixture and, therefore, tested together.

## Data Availability

The original contributions presented in this study are included in the article/Appendix A. Further inquiries can be directed to the corresponding author.

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
