# Peer review of "Antiprotozoal Aminosteroids from Pachysandra terminalis"

_molecules, 2025, doi:10.3390/molecules30051093_

Round 1
Reviewer 1 Report
Comments and Suggestions for Authors
The manuscript studies antiparasitic activity against the hemoflagellate protozoan Trypanosoma brucei, the etiological agent of HAT and Plasmodium falciparum. Since this is a scientific effort to develop therapeutic alternatives for NTDs, it will arouse the interest of members of the scientific community who are dedicated to the search for solutions for these severe infectious diseases.
Please find in the attached PDF the comments made on some points of the manuscript that, in our opinion, can improve the fluidity and clarity of the text.
General comments:
1. Abstract:
The authors should highlight, in the abstract and also in the manuscript body, the importance of fractionation guided by the results of biological evaluation, which is an essential tool in the area of medicinal chemistry of natural products.
2. The plant species studied and the aminosterols present in the plant matrices have different studies on their antiproliferative activities in different tumor cell lines (and are even more studied in this type of biological model). Since the kinetics of cell proliferation in tumors and unicellular parasites are similar, the authors should include a comment on this in the introduction, which would also support the study carried out by the group.
See: https://doi.org/10.3390/molecules21101283; https://doi.org/10.1016/j.phytol.2021.07.014; https://doi.org/10.1021/np300207c.
3. In the last paragraph of the introduction, the authors may discuss the differences and similarities of possible biochemical targets in the two parasites to justify the study of antiparasitic activity, specifically in T. brucei and P. falciparum. The following work may help: doi: 10.1016/j.pt.2011.01.004. PMID: 21345732;
4. The discussions involving the structural determination of the isolated compounds are very enlightening and agree with works in the area. However, due to the interdisciplinary aspect of the present work, this reviewer suggests that the authors include a figure that illustrates the interpretation of NMR spectra [mainly using as an example one (or more) of the compounds isolated for the first time].
5. lines 246-262: This experimental observation deserves greater emphasis and discussion (perhaps with a descriptive Figure), as it will provide information for other authors working on similar themes.
6. lines 319-321 (extraction of plant material): I did not understand what was stated here by the authors. I would ask for more details on this procedure, since a percolator was not used. What is the volume of the extracting solvent, and what flow rate was used? In my opinion, important information are missing on this topic.
7. lines 495-498 (Biological assessments) Although the group has already carried out the protocol for evaluating the biological activity of the two protozoa in the cited article [ref. 35], it is essential to include here a complete and more detailed description of the protocol for the biological tests performed (strains used, as well as their origins, reagents and culture media, the concentration ranges tested for each compound, in addition to the methodology used for the statistical treatment of the observed biological data, which should also be referenced). A lot is missing here in this M&M section of the biological tests. PI noticed that some information is in the supplementary material, but in my opinion, it should be described in the body of the manuscript.
The manuscript is interesting and presents very relevant data. Therefore, after making the suggested modifications, it can be accepted for publication in Molecules.
PS: The PDF of the manuscript has been uploaded, and the considerations listed above are marked in the text along with the comments that were included.

Author Response
Reviewer 1
The manuscript studies antiparasitic activity against the hemoflagellate protozoan Trypanosoma brucei, the etiological agent of HAT and Plasmodium falciparum. Since this is a scientific effort to develop therapeutic alternatives for NTDs, it will arouse the interest of members of the scientific community who are dedicated to the search for solutions for these severe infectious diseases.
Please find in the attached PDF the comments made on some points of the manuscript that, in our opinion, can improve the fluidity and clarity of the text.
General comments:
- Abstract:
The authors should highlight, in the abstract and also in the manuscript body, the importance of fractionation guided by the results of biological evaluation, which is an essential tool in the area of medicinal chemistry of natural products.
Reply: We agree with this from a general point of view. In the present work, however, we did not strictly follow an activity-guided protocol since we intended to obtain aminosteroids of varying activity, in order to conduct structure-activty relationship studies at a later stage. We have added short statements to clarify this: (Abstract): “…In the present study, the alkaloid-enriched fraction obtained from a 75% ethanol extract of aerial parts was separated to isolate a chemically diverse array of Pachysandra alkaloids for for assessment of their antiprotozoal activity and later structure-activity studies. This work yielded a new megastigmane alkaloid…”; (Introduction, sentence before last): “Therefore, the present study was aimed at increasing the chemical diversity of known antiprotozoal aminosteroid alkaloids by isolating such compounds from P. terminalis and evaluating them as potential hits or leads with antitrypanosomal and/or antiplasmodial activity.”
- The plant species studied and the aminosterols present in the plant matrices have different studies on their antiproliferative activities in different tumor cell lines (and are even more studied in this type of biological model). Since the kinetics of cell proliferation in tumors and unicellular parasites are similar, the authors should include a comment on this in the introduction, which would also support the study carried out by the group.
See: https://doi.org/10.3390/molecules21101283; https://doi.org/10.1016/j.phytol.2021.07.014; https://doi.org/10.1021/np300207c.
Reply: Of course, we agree with the reviewer and have added the following sentence in the introduction: “Various Pachysandra alkaloids have been reported to exert conspicuous biological activity, particularly anticancer effects [references].
- In the last paragraph of the introduction, the authors may discuss the differences and similarities of possible biochemical targets in the two parasites to justify the study of antiparasitic activity, specifically in T. brucei and P. falciparum. The following work may help: doi: 10.1016/j.pt.2011.01.004. PMID: 21345732;
Reply: We thank the reviewer for this interesting remark. We have added a short statement at the end of the introduction taking into account that both parasites under study may have certain commonalities and added the suggested reference: “Even though parasites of genera Trypanosoma and Plasmodium belong to rather remote branches of the Eukaryote tree of life, they may share certain strategies of transmission and survival [reference] as well as biochemical targets and susceptibility towards antiproto-zoal agents. It will hence also be interesting to compare their susceptibility towards Pachysandra alkaloids.”
We also added a short comparison of the different parasites’ susceptibility for the isolated compounds (last part of section 2.3).
- The discussions involving the structural determination of the isolated compounds are very enlightening and agree with works in the area. However, due to the interdisciplinary aspect of the present work, this reviewer suggests that the authors include a figure that illustrates the interpretation of NMR spectra [mainly using as an example one (or more) of the compounds isolated for the first time].
Reply: We thank the reviewer for this constructive suggestion. We have added a figure (now Figure 3) exemplifying the assignment of stereochemistry at position 4 in compound 7.
- lines 246-262: This experimental observation deserves greater emphasis and discussion (perhaps with a descriptive Figure), as it will provide information for other authors working on similar themes.
Reply: Again, we thank the reviewer for the suggestion to add further illustrative material. We have added a Figure (now Figure 4) to illustrate the formation of compound 9 from compound 19 during the prep. HPLC separation.
- lines 319-321 (extraction of plant material): I did not understand what was stated here by the authors. I would ask for more details on this procedure, since a percolator was not used. What is the volume of the extracting solvent, and what flow rate was used? In my opinion, important information are missing on this topic.
Reply: We have clarified this statement and added the amount of solvent as well as the flow rate. It now reads: “The dried and crushed aerial parts of P. terminalis (1043 g) were extracted through percolation using a glass chromatography tube (92 x 7 cm) as percolator and aqueous ethanol (75%) as extracting solvent. The plant material was soaked with the solvent in the percolator overnight (no flow) and then extracted with 1000 mL of solvent (flow rate approx. 8 mL/min). This procedure was carried out three subsequent times.” We hope it is clearer this way.
- lines 495-498 (Biological assessments) Although the group has already carried out the protocol for evaluating the biological activity of the two protozoa in the cited article [ref. 35], it is essential to include here a complete and more detailed description of the protocol for the biological tests performed (strains used, as well as their origins, reagents and culture media, the concentration ranges tested for each compound, in addition to the methodology used for the statistical treatment of the observed biological data, which should also be referenced). A lot is missing here in this M&M section of the biological tests. PI noticed that some information is in the supplementary material, but in my opinion, it should be described in the body of the manuscript.
Reply: The requested detailed information has been added.
The manuscript is interesting and presents very relevant data. Therefore, after making the suggested modifications, it can be accepted for publication in Molecules.
PS: The PDF of the manuscript has been uploaded, and the considerations listed above are marked in the text along with the comments that were included.
We thank the reviewer for the constructive assessment which has helped us improve our work!

Reviewer 2 Report
Comments and Suggestions for Authors
This manuscript is focused on the isolation, structural characterisation and evaluation of the aminosteroids derived from Pachysandra terminalis, an evergreen subshrub. Based on previous research, this plant has been confirmed as a rich source of biologically active alkaloids with promising activity against protozoan parasites. These compounds were tested against two harmful protozoan parasites: Trypanosoma brucei rhodesiense, the causative agent of human African trypanosomiasis (HAT), and Plasmodium falciparum, responsible for malaria. In this work, a highly complex 75% ethanolic extract of Pachysandra terminalis yielded 20 alkaloids, highlighting the plant's exceptional chemical diversity. Structural characterisation of the isolated compounds was achieved through detailed NMR and LC/MS analyses, which highlighted their novelty. In particular, the alkaloid-enriched fraction and individual compounds exhibited potent antiprotozoal activity. 3α,4α-diapachysanaximine A was the most active, with IC50 values in the low micromolar range against both Plasmodium falciparum and Trypanosoma brucei rhodesiense. The results of this work underline the potential of P. terminalis aminosteroids as promising antiprotozoal agents. Efforts are currently underway to isolate 3α,4α-diapachysanaximine A in larger quantities to facilitate in vivo assays, paving the way for further investigation of its therapeutic potential.
I will recommend the acceptance after minor revisions (listed below).
- Expand the introduction to better contextualise aminosteroids in relation to existing antiprotozoal therapies. Highlight the weaknesses of current treatments and the role of natural products in addressing drug resistance. Include a section mentioning the importance of multidisciplinary approaches that integrate structural biology, medicinal chemistry and enzymology. For instance, studies exploiting high-resolution crystal structures of enzyme-inhibitor complexes have paved the way for rational drug design, showing how detailed structural information can guide the development of innovative and potent therapies (citation suggested: DOI 10.1107/S2059798320004891).
2. If structural and biological data allow, the authors should elaborate on the structure-activity relationships (SAR) of the isolated compounds. It would be useful to discuss how specific functional groups may influence the bioactivity and selectivity observed against Plasmodium falciparum and Trypanosoma brucei. If sufficient data are not available, this aspect could be addressed in future studies.
Author Response
Reviewer 2
This manuscript is focused on the isolation, structural characterisation and evaluation of the aminosteroids derived from Pachysandra terminalis, an evergreen subshrub. Based on previous research, this plant has been confirmed as a rich source of biologically active alkaloids with promising activity against protozoan parasites. These compounds were tested against two harmful protozoan parasites: Trypanosoma brucei rhodesiense, the causative agent of human African trypanosomiasis (HAT), and Plasmodium falciparum, responsible for malaria. In this work, a highly complex 75% ethanolic extract of Pachysandra terminalis yielded 20 alkaloids, highlighting the plant's exceptional chemical diversity. Structural characterisation of the isolated compounds was achieved through detailed NMR and LC/MS analyses, which highlighted their novelty. In particular, the alkaloid-enriched fraction and individual compounds exhibited potent antiprotozoal activity. 3α,4α-diapachysanaximine A was the most active, with IC50 values in the low micromolar range against both Plasmodium falciparum and Trypanosoma brucei rhodesiense. The results of this work underline the potential of P. terminalis aminosteroids as promising antiprotozoal agents. Efforts are currently underway to isolate 3α,4α-diapachysanaximine A in larger quantities to facilitate in vivo assays, paving the way for further investigation of its therapeutic potential.
I will recommend the acceptance after minor revisions (listed below).
- Expand the introduction to better contextualise aminosteroids in relation to existing antiprotozoal therapies. Highlight the weaknesses of current treatments and the role of natural products in addressing drug resistance. Include a section mentioning the importance of multidisciplinary approaches that integrate structural biology, medicinal chemistry and enzymology. For instance, studies exploiting high-resolution crystal structures of enzyme-inhibitor complexes have paved the way for rational drug design, showing how detailed structural information can guide the development of innovative and potent therapies (citation suggested: DOI 10.1107/S2059798320004891).
Reply: A short general statement on the importance of natural products in the search for new antiprotozoal drugs was added in the introduction (first paragraph). Also, we added a statement that aminosteroids, according to some investigations of our group, may possess a new mechanism of action and are hence of particular interest (second paragraph).
We would rather not add further sections on general approaches or discuss this in too much detail here since it will only distract from the actual topic. We would also rather not add a citation of a publication on a particular parasite enzyme since PTR1 this is not the topic of this present work.
- If structural and biological data allow, the authors should elaborate on the structure-activity relationships (SAR) of the isolated compounds. It would be useful to discuss how specific functional groups may influence the bioactivity and selectivity observed against Plasmodium falciparumand Trypanosoma brucei. If sufficient data are not available, this aspect could be addressed in future studies.
Reply: It is indeed envisaged to address detailed (Q)SAR (together with compounds from our previous studies and further ongoing ones) in a future study. A short statement has been added at the end of the conclusions: “Investigations on structure-activity-relationships of the isolated Pachysandra alkaloids in comparison with those obtained from Buxus and Holarrhena species are under way.”
We thank the reviewer for the constructive assessment which has helped us improve our work!

Reviewer 3 Report
Comments and Suggestions for Authors
This manuscript describes the extraction, fractionation, purification and identification of compounds from Pachysandra terminalis extracts. The work show that this plant has many compounds with anti-protozoal activity, and can be a source of lead molecules for new drug formulations against protozoan parasites. The chemical part of the manuscript is clearly described, but the in vitro bioactivities are not, this needs to be completed to provide the best understanding of the manuscript-
The introduction is quite brief, but that is okay.
Results:
The data in Table 1 and Table 4 show interesting and promising results, but their interpretation is difficult given the lack of data on the in vitro antiprotozoal methods and incubation time used. These parameters affect the IC50 values, so they must be known. Please add this information.
Why did the authors use the L6 cell line for in vitro cytotoxicity? Which was the objective, or what did the authors wanted to demonstrate or observe with the cytotoxicity results? This point is unclear. Please clarify. Discussion of results should be implemented, especially when correlating toxicity results.
Methods, section 4.6 should contain the minimum essential to be comprehensive. The description should contain the origin of microorganisms and cells, the essential steps of the used method (name of the method, main reagents, main steps…).
For example, in cytotoxic test authors must indicate, the conditions of culture, how many hours were the cells exposed to the compounds. The same for the in vitro antiprotozoal activity.
In methods, please add a section “Data and statistical analysis” and refer to the statistical tests used.
The supplementary material is well documented and complements the reading of the article.
Comments on the Quality of English LanguageEnglish only needs minor corrections (e.g. verb tenses)
Author Response
Reviewer 3
This manuscript describes the extraction, fractionation, purification and identification of compounds from Pachysandra terminalis extracts. The work show that this plant has many compounds with anti-protozoal activity, and can be a source of lead molecules for new drug formulations against protozoan parasites. The chemical part of the manuscript is clearly described, but the in vitro bioactivities are not, this needs to be completed to provide the best understanding of the manuscript-
The introduction is quite brief, but that is okay.
Results:
The data in Table 1 and Table 4 show interesting and promising results, but their interpretation is difficult given the lack of data on the in vitro antiprotozoal methods and incubation time used. These parameters affect the IC50 values, so they must be known. Please add this information.
Reply: Detailed information on the bioassays has been added to the Materials and Methods section.
Why did the authors use the L6 cell line for in vitro cytotoxicity? Which was the objective, or what did the authors wanted to demonstrate or observe with the cytotoxicity results? This point is unclear. Please clarify. Discussion of results should be implemented, especially when correlating toxicity results.
Reply: The L6 cell line is a mammalian cell line and has been used very frequently to represent mammalian cytotoxicity in comparison with antiprotozoal activity. Originally, this cell line was chosen at Swiss TPH because it also serves as host cell for intracellular forms of Trypanosoma cruzi. But since it has been used so often over the decades, the cytotoxicity results of very many different compounds tested against this cell line can be compared very well. A short statement clarifying this fact has been added to the first Results section (2.1).
Methods, section 4.6 should contain the minimum essential to be comprehensive. The description should contain the origin of microorganisms and cells, the essential steps of the used method (name of the method, main reagents, main steps…). For example, in cytotoxic test authors must indicate, the conditions of culture, how many hours were the cells exposed to the compounds. The same for the in vitro antiprotozoal activity.
Reply: All this information has been added to the Materials and Methods section, section 4.6.
In methods, please add a section “Data and statistical analysis” and refer to the statistical tests used.
Reply: The requested explanation has been added at the end of section 4.6: “Bioassay data and statistical methods: In all cases, the readout data were plotted in Microsoft Excel. Half-maximal inhibitory concentrations (IC50 values) were calculated from the sigmoidal dose-response curves by linear regression [44]. Two independent replicates of the assay were performed in all cases and the results expressed as arithmetic mean ± deviation from the mean.”
The supplementary material is well documented and complements the reading of the article.
We thank the reviewer for the constructive assessment which has helped us improve our work!

Reviewer 4 Report
Comments and Suggestions for Authors
The manuscript "Antiprotozoal Aminosteroids from Pachysandra terminalis" reports the discovery of a new megastigmane alkaloid (compound 1) and seven new aminosteroids (compounds 2, 7, 16, 17, 18, 19, and 20). The authors also characterized two artifacts (compounds 6 and 9) formed during the isolation process. Following extraction, the compounds were evaluated for activity against Trypanosoma brucei rhodesiense and Plasmodium falciparum, as well as toxicity in L6 cells (rat skeletal myoblasts) to determine the Selectivity Index (SI).
The isolated compounds, including the two artifacts, were characterized using LC/MS and detailed NMR spectroscopy. Finally, the isolated compounds were tested against Trypanosoma brucei rhodesiense and Plasmodium falciparum, and the SI was calculated for all compounds.
The authors conclude by mentioning that the isolation of 3α,4α-diapachysanaximine A in larger quantities for in vivo assays is in progress.
Minor points for revision:
- The manuscript should clarify the use of LC/MS in determining the structural formulas of the compounds, detailing the specific methodology employed.
- Table 4 would be clearer if it presented concentrations using only one unit.
- To enhance the study, I suggest performing statistical analysis to compare the IC50 values of all isolated compounds and to initiate a discussion of structure-activity relationships (SAR). For example, the superior activity of artifact compound 6 compared to all other isolated compounds should be highlighted in the discussion and conclusion.
Author Response
Reviewer 4
The manuscript "Antiprotozoal Aminosteroids from Pachysandra terminalis" reports the discovery of a new megastigmane alkaloid (compound 1) and seven new aminosteroids (compounds 2, 7, 16, 17, 18, 19, and 20). The authors also characterized two artifacts (compounds 6 and 9) formed during the isolation process. Following extraction, the compounds were evaluated for activity against Trypanosoma brucei rhodesiense and Plasmodium falciparum, as well as toxicity in L6 cells (rat skeletal myoblasts) to determine the Selectivity Index (SI).
The isolated compounds, including the two artifacts, were characterized using LC/MS and detailed NMR spectroscopy. Finally, the isolated compounds were tested against Trypanosoma brucei rhodesiense and Plasmodium falciparum, and the SI was calculated for all compounds.
The authors conclude by mentioning that the isolation of 3α,4α-diapachysanaximine A in larger quantities for in vivo assays is in progress.
Minor points for revision:
- The manuscript should clarify the use of LC/MS in determining the structural formulas of the compounds, detailing the specific methodology employed.
Reply: Indeed, ESI mass spectral fragmentation yields very helpful information for structure elucidation of these compounds. We have added a statement acknowledging this and citing two references describing this (Section 2.2., second paragraph). “Also, in all cases of aminosteroids, the mass spectral data, including the characteristic fragmentation pattern as previously described [refs] supported the structural assignments.”
- Table 4 would be clearer if it presented concentrations using only one unit.
Reply: We prefer to keep both units because the values in µg/mL (with deviations) were the raw data obtained from the bioassays while the µM values (only means) were calculated but make comparison between the different compounds easier.
- To enhance the study, I suggest performing statistical analysis to compare the IC50 values of all isolated compounds and to initiate a discussion of structure-activity relationships (SAR). For example, the superior activity of artifact compound 6 compared to all other isolated compounds should be highlighted in the discussion and conclusion.
Reply: We find it interesting to present a comparison of the biological activities and have added a brief discussion on this topic at the end of section 2.3.
A detailed investigation of structure-activity relationships of the present compounds in comparison with those obtained from Buxus and Holarrhena species is in progress. We therefore added a short statement at the end of the conclusions to envisage these ongoing studies. “Investigations on structure-activity-relationships of the isolated Pachysandra alkaloids in comparison with those obtained from Buxus and Holarrhena species are under way.”
We thank the reviewer for the constructive assessment which has helped us improve our work!

Round 2
Reviewer 1 Report
Comments and Suggestions for Authors
I want the authors to review the figures in the supplementary material. There are some inconsistencies. The list of figures presented at the beginning of the session goes up to Fig. S195, but the last figure that appears is S169. Please review and correct it.
Author Response
We are very sorry, but obviously the reviewer received an incomplete Supplementary file.
We have checked that our submitted document contains 195 supplementary figures. We upload the file again and hope it will be received correctly.
We thank the reviewer for the scrutinous work!